# Few-Shot Learner Generalizes Across AI-Generated Image Detection

**Shiyu Wu** [1 2 3]   **Jing Liu** [1 3]   **Jing Li** [4]   **Yequan Wang** [2]

## Abstract

Current fake image detectors trained on large synthetic image datasets perform satisfactorily on limited studied generative models. However, these detectors suffer a notable performance decline over unseen models. Besides, collecting adequate training data from online generative models is often expensive or infeasible. To overcome these issues, we propose Few-Shot Detector (FSD), a novel AI-generated image detector which learns a specialized metric space for effectively distinguishing unseen fake images using very few samples. Experiments show that FSD achieves state-of-the-art performance by $+11.6\%$ average accuracy on the GenImage dataset with only $10$ additional samples. More importantly, our method is better capable of capturing the intra-category commonality in unseen images without further training. Our code is available at https://github.com/teheperinko541/Few-Shot-AIGI-Detector.

## 1. Introduction

The development of generative models has led to significant strides in synthesizing photorealistic images, making it much more difficult to distinguish AI-generated images from real ones. Over the past few years, diffusion-based models (Ho et al., 2020) have exceeded GANs (Goodfellow et al., 2014) and become the mainstream paradigm of image generation, due to their exceptional ability to synthesize high-quality images. With the rapid release of open-source models and web APIs, users can easily create vivid images from brief textual descriptions. However, the increasing abuse of deepfake technology has dramatically influenced human society and sparked unprecedented concerns on the credibility of online information.

To curb the spread of AI-generated images used for malicious purposes, there is a growing urgency to develop detection methods capable of distinguishing fake images from authentic ones. Early works (Zhang et al., 2019b; Chai et al., 2020) focus on identifying artifacts in synthetic images, which have proved to be particularly useful for detecting GAN-generated images. With the improvement of synthetic image datasets (Zhu et al., 2023; Boychev & Cholakov, 2024), detecting fake images from seen generative models is no longer a difficult task. Thus, a novel challenge arises in developing classifiers that generalize across unseen models. To address this problem, some studies (Ojha et al., 2023; Liu et al., 2024) adopt a large vision-language model to analyze images from a semantic perspective, while others (Wang et al., 2023; Luo et al., 2024) leverage the reconstruction ability of diffusion models.

Another difficulty is the acquisition of training data. It is often expensive or infeasible to collect enough images from closed-source models such as DALL-E (Ramesh et al., 2021) and Midjourney (mid, 2022). This is due to their service prices and access restrictions. However, most existing fake image detectors (Wang et al., 2023; Ojha et al., 2023) require large amounts of training data to achieve good generalization, and fine-tuning without adequate data often results in overfitting. Furthermore, the ongoing release and updating of generative models pose a significant challenge for existing detection systems to keep up.

After gaining a deep insight into these limitations, we propose an innovative approach based on few-shot learning to overcome them. A robust model is also developed to validate the efficacy of our solution. We first point out that current AI-generated image detection is a domain generalization task. Previous studies have been dedicated to discovering a universal indicator effective for detecting diverse fake images. However, they overlook the significant distinctions among data from different domains. We observe that images from unseen domains can actually be obtained in many real-world scenarios. Based on this fact, the complicated task can be transformed into a more tractable one, called few-shot classification, through limited samples from unseen domains. Thus, we can extract rich domain-specific

[1]Institute of Automation, Chinese Academy of Sciences, Beijing, China [2]Beijing Academy of Artificial Intelligence, Beijing, China [3]University of Chinese Academy of Sciences, Beijing, China [4]Harbin Institute of Technology, Shenzhen, China. Correspondence to: Yequan Wang <tshwangyequan@gmail.com>, Jing Liu <jliu@nlpr.ia.ac.cn>.

*Proceedings of the 42nd International Conference on Machine Learning*, Vancouver, Canada. PMLR 267, 2025. Copyright 2025 by the author(s).

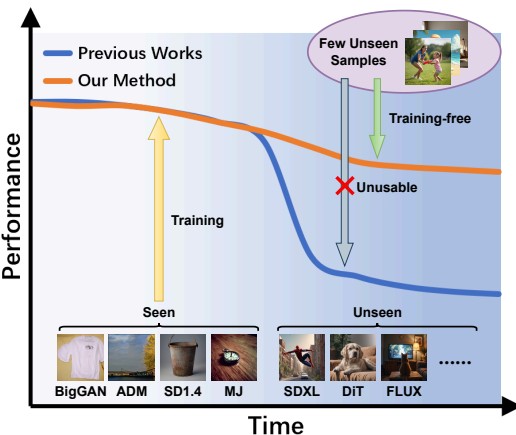

*Figure 1.* Challenge facing previous works and our solution. Most classifiers exhibit significant performance degradation over unseen data. We address this challenge by introducing a few-shot strategy which is able to make full use of the limited unseen samples.

features from the given samples and use them to generalize across the unseen data, as shown in Figure 1.

Based on the above analysis, we propose Few-Shot Detector (FSD), a novel method which is able to detect fake images with few given samples. In the few-shot classification task, synthetic images from different generative models are classified into separate categories, whereas the real images are classified into a single category. The objective is to develop a classifier capable of generalizing to unseen classes that are not in the training set, with only a limited number of examples from them. To solve this problem, FSD utilizes the Prototypical Network (Snell et al., 2017), which learns a metric space to effectively handle unseen data. The given samples are used to compute the prototypical representations of different classes. The test image is then classified using the nearest-neighbor method by comparing it against these representations. Thus, FSD can automatically adjust for bias in previously unseen classes without additional training or fine-tuning.

Experiments show that FSD achieves notable improvements over the state-of-the-art method by $+11.6\%$ average accuracy on the GenImage dataset. We observe that only 10 samples from the tested classes can significantly improve the detection performance, which highlights the superiority and practicality of FSD. We also show that our approach is better equipped to keep pace with the rapid advancements in image generation, thereby demonstrating the effectiveness of categorizing generated images by their sources.

Our contributions are summarized as follows:

- To the best of our knowledge, we are the first to reconceptualize AI-generated image detection as a few-shot classification task, bringing this task closer to real-world applications.

- We introduce an innovative synthetic image detection method, named Few-Shot Detector (FSD). Our approach is able to utilize a few samples from unseen domains to achieve better detection performance.

- Our experiments demonstrate the generalization ability of FSD, which significantly outperforms current state-of-the-art methods and can easily deal with new generative models without further training.

## 2. Related Work

### 2.1. Image Synthesis

Generative Adversarial Networks (GANs) (Goodfellow et al., 2014) have revolutionized the field of generative modeling over the last decade. By simultaneously training a generator and a discriminator in an adversarial manner, GANs can generate samples of high resolution and good quality (Brock et al., 2019; Karras et al., 2020). However, they are prone to mode collapse, which makes it difficult for them to fully capture the data distribution. An alternative approach is likelihood-based modeling, which estimates the parameters of data distributions by maximizing the likelihood function. Based on this methodology, variational autoencoders (VAEs) (Kingma & Welling, 2014) ensure a smooth latent space that facilitates interpolation and exploration of new samples. In contrast, normalizing flow models (Papamakarios et al., 2021) enable exact log-likelihood computation, allowing them to capture more complex data distributions. However, neither of them can match GANs in high-quality image generation.

Diffusion Models (Ho et al., 2020; Song et al., 2021) have recently achieved remarkable performance in image synthesis. By reversing a gradual noising process, diffusion models can generate images from random noise through successive denoising steps. Ho & Salimans (2022) introduce classifier-free guidance which enables generating images from text descriptions, eliminating the need for training a separate classifier. Another significant advancement is the Latent Diffusion Model (LDM) (Rombach et al., 2022), which applies the diffusion process in latent space with a powerful autoencoder. This approach greatly enhances the visual fidelity of synthetic images while substantially reducing the required computational resources. Leveraging vast amounts of training data, large-scale latent diffusion models (Podell et al., 2024; Esser et al., 2024) have emerged as the leading technique in image generation and the main source of synthetic images at present.

### 2.2. AI-generated Image Detection

**Model-agnostic detection method.** Learning-based methods have been applied to detecting fake images from GANs

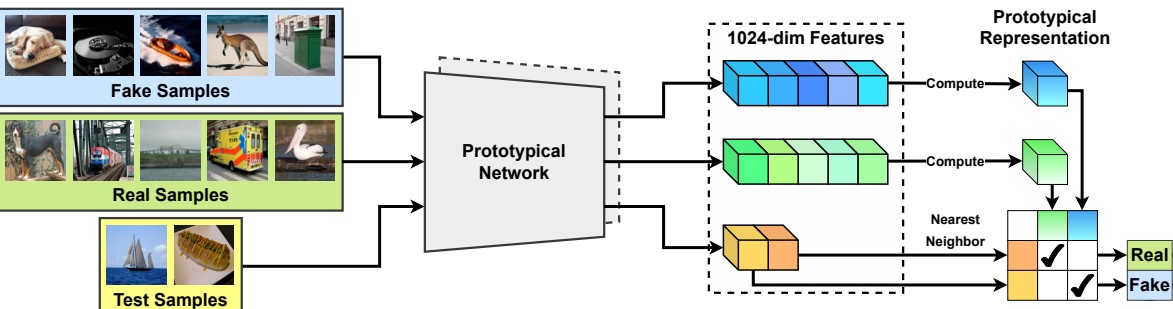

*Figure 2.* Detection pipeline of Few-Shot Detector. FSD first maps the real and fake images from the support set to the metric space by the Prototypical Network and calculates the representation of each type. A test image is then mapped into the same space and classified according to the nearest-neighbor principle.

(Zhang et al., 2019b; Qian et al., 2020; Chai et al., 2020). They treat synthetic image detection as a binary classification task, aiming to find synthetic clues in generated images. Numerous studies (Zhang et al., 2019a; Gragnaniello et al., 2021) have found that classifiers often fail to generalize across unseen generative models, bringing generalization ability to the primary research objective. Wang et al. (2020) reveal that detectors for CNN-generated images can be surprisingly generalizable due to the common systematic flaws shared among CNNs. Based on this work, Liu et al. (2023) analyze multi-view features to learn a robust representation, while Tan et al. (2024b) focus on high-frequency information to capture source-agnostic features.

However, these common cues disappear when it comes to diffusion models, leading to the poor performance of earlier methods when applied to them. Therefore, more attention has been paid to semantic level. For instance, Ojha et al. (2023) prove that the informative image features from a pretrained CLIP model (Radford et al., 2021) can be successfully used for detecting fake images. Other studies (Liu et al., 2024; Tan et al., 2024a) deeply integrate textual information with images, thereby further enhancing the generalization capability. Moreover, a patch-based detection method (Chen et al., 2024) extracts features from a single patch rather than the entire image, which significantly accelerates the detection speed.

**Diffusion-based detection method.** The reconstruction capability of open-source diffusion models has opened a new technological pathway. DIRE (Wang et al., 2023) first uses the reconstruction loss to train a classifier, based on the observation that synthetic images can be reconstructed with higher fidelity compared to real ones. SeDID (Ma et al., 2023) analyzes the loss errors between noised and denoised features at specific steps of processing, while LARE$^2$ (Luo et al., 2024) focuses on the single-step reconstruction result in latent space. Both of them improve the detection speed by not fully completing the entire reconstruction process. FakeInversion (Cazenavette et al., 2024) employs

BLIP (Li et al., 2022) to obtain a text description of the image for better reconstruction, which is capable of detecting unseen high-fidelity generated images. AEROBLADE (Ricker et al., 2024) evaluates autoencoder (AE) in different latent diffusion models and presents a training-free method for detecting fake images based on AE reconstruction errors. Although these approaches have achieved great success on open-source generative models, they still face challenges in detecting synthetic images from various inaccessible models due to the substantial differences among them.

## 3. Method

In this section, we first describe the few-shot classification task for synthetic image detection. Then we elaborate on our novel representation, namely Few-Shot Detector (FSD), as illustrated in Figure 2.

### 3.1. Few-shot Synthetic Image Detection

Few-shot classification is a specialized machine learning task, where the classifier is challenged to generalize to entirely new classes not present in the training set, using only a limited number of examples from these novel classes. Different from end-to-end detection methods that aggregate all synthetic images produced by different generative models into a single fake class, we classify these images based on their generators. This means that we consider the productions of each model as an individual class. Conversely, authentic images continue to be grouped into a single class. During detection, the support set comprises labeled images aiding in the classification task, while the query set contains unlabeled images to be classified. The objective of this task is to determine the category of each image in the query set based on the support set.

In detail, we are given a support set $\mathcal{S} = \mathcal{S}_1 \cup \mathcal{S}_2 \cup \cdots \cup \mathcal{S}_N$, which comprises samples from $N$ new classes. Each subset $\mathcal{S}_i = \{(\mathbf{x}_{i1}, y_{i1}), (\mathbf{x}_{i2}, y_{i2}), \ldots, (\mathbf{x}_{iK}, y_{iK})\}$ denotes the collection of images from class $i$. $K$ is the number of sam-

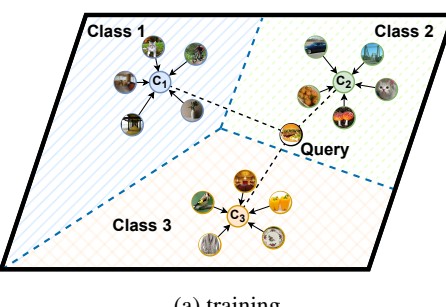

(a) training

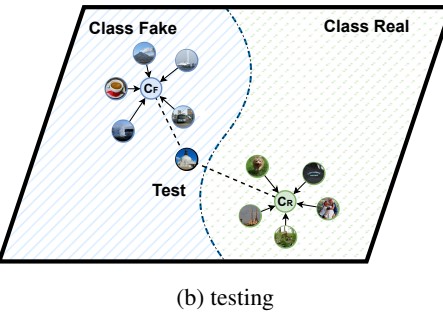

(b) testing

*Figure 3.* Prototypical calculation. During training, the metric space is learned by minimizing the distances between query samples and their corresponding centers. During testing, the test image is only classified between real and fake.

ples in each class. $\mathbf{x}_{ij} \in \mathbb{R}^D$ is an image of $D$ dimensions and $y_{ij} = i$ indicates its corresponding label. Each unlabeled image in the query set will be assigned to one of the $N$ classes. This is the so-called $N$-way $K$-shot task.

### 3.2. Few-Shot Detector

Our FSD is based on the Prototypical Network (Snell et al., 2017), an effective method for addressing the few-shot question above. The Prototypical Network aims to learn a metric space in which samples from the same class are close in distance, while samples from different classes are relatively distant from each other. Thus, we can represent each class by a single vector, known as the prototypical representation. For each sample in the query set, classification is conducted by simply finding the nearest representation to it.

In detail, as depicted in Figure 2, FSD adopts a neural network $f_\phi : \mathbb{R}^D \rightarrow \mathbb{R}^M$ to compute the $M$-dimensional vector in the metric space of an image, where $\phi$ is the learnable parameters. The prototypical representation $\mathbf{c}_i$ for class $i$ is defined as the average of the embedded vectors derived from its corresponding support set $\mathcal{S}_i$:

$$\mathbf{c}_i = \frac{1}{|\mathcal{S}_i|} \sum_{\mathbf{x}_j \in \mathcal{S}_i} f_\phi(\mathbf{x}_j). \quad (1)$$

After gaining all the $N$ prototypical representations and the vector $f_\phi(\mathbf{x}_q)$ of a test image $\mathbf{x}_q$ in the query set, we

---

**Algorithm 1** Prototypical Loss Computation Algorithm

**Input:** Training set $\mathcal{D} = \mathcal{D}_1 \cup \mathcal{D}_2 \cup \cdots \mathcal{D}_N$, where $\mathcal{D}_k$ denotes the subset of class $k$; number of training class $N_c$; number of support examples per class $N_s$; number of query examples per class $N_q$.
**Output:** training loss $J$.
$V \leftarrow$ randomly select $N_c$ classes from $\{1, 2, \ldots, N\}$
**for** $k$ in $V$ **do**
    $S_k \leftarrow$ randomly select $N_s$ samples from $\mathcal{D}_k$
    $Q_k \leftarrow$ randomly select $N_q$ samples from $\mathcal{D}_k/S_k$
    $\mathbf{c}_k \leftarrow \frac{1}{N_s} \sum_{\mathbf{x}_i \in S_k} f_\phi(\mathbf{x}_i)$
**end for**
$J \leftarrow 0$
**for** $k$ in $V$ **do**
    $J \leftarrow J - \frac{1}{N_c N_q} \sum_{\mathbf{x}_j \in Q_k} \log \text{Softmax}(-d(f_\phi(\mathbf{x}_j), \mathbf{c}_k))$
**end for**
**Return** $J$

---

compute the distances between the test vector and all the representations $d(f_\phi(\mathbf{x}_q), \mathbf{c}_i)$, using a distance function $d : \mathbb{R}^M \times \mathbb{R}^M \rightarrow \mathbb{R}_0^+$. The probability that the query sample $\mathbf{x}_q$ belongs to class $i$ is determined by:

$$p(y = i|\mathbf{x}_q) = \text{Softmax}_{1 \leq i \leq N}(-d(f_\phi(\mathbf{x}_q), \mathbf{c}_i)). \quad (2)$$

The image $\mathbf{x}_q$ is assigned to the class with the maximum probability, or the nearest prototypical representation to it.

As shown in Figure 3(a), during each training step, $N_c$ classes are randomly selected from the training set. For each selected class $i$, we randomly choose $N_s$ samples from it as the support set $\mathcal{S}_i$ and another $N_q$ samples as the query set $\mathcal{Q}_i$. We compute the $N_c$ prototypical representations according to Equation (1). The Prototypical Network outputs a probability distribution over the $N_c$ classes for each sample in the query set. The optimization target is to minimize the negative log-probability $J(\phi)$. Pseudo code to compute the loss $J(\phi)$ is provided in Algorithm 1.

### 3.3. Zero-shot Classification

We propose a zero-shot detection approach to verify the applicability of FSD in real-world settings. In the zero-shot task, there are no additional samples from the test classes. Instead, we are provided with a metadata vector for each class in the training set to categorize the test images. We put considerable emphasis on this task, since acquiring samples from all potential generative models can be impractical. The zero-shot results can represent the performance of our model in general scenarios.

In this work, we simply define the metadata vector as the prototypical representation derived from an extensive collection of samples. That is, we randomly select a large number of images from each class in the training set and then com-

*Table 1.* Comparison of our FSD with previous studies. We train several classifiers on different subsets and report the accuracy for each unseen test subset. The best results are highlighted in boldface, while the second-optimal results are marked with underline.

| Method | Test Subset | | | | | | Avg |
|---|---|---|---|---|---|---|---|
| | Midjourney | GLIDE | ADM | SD | VQDM | BigGAN | Acc(%) |
| Spec (Zhang et al., 2019b) | 50.0 | 64.7 | 52.8 | 56.1 | 56.5 | 63.0 | 57.2 |
| CNNSpot (Wang et al., 2020) | 52.8 | 73.3 | 55.0 | 55.9 | 54.4 | 66.2 | 59.6 |
| F3Net (Qian et al., 2020) | 50.1 | 52.5 | 66.4 | 57.0 | 59.4 | 50.4 | 56.0 |
| GramNet (Liu et al., 2020) | 51.3 | 62.6 | 53.8 | 56.8 | 52.2 | 57.2 | 55.7 |
| DIRE (Wang et al., 2023) | 57.9 | 68.2 | 57.3 | 58.2 | 59.6 | 50.8 | 58.7 |
| LARE2 (Luo et al., 2024) | 62.7 | 80.2 | 63.5 | 79.6 | **76.9** | 72.0 | 72.5 |
| FSD (zero-shot) | 75.1 | 93.9 | 74.1 | 88.0 | 69.1 | 62.1 | 77.1 |
| FSD (10-shot) | **80.9** | **97.1** | **79.2** | **88.8** | 76.2 | **82.2** | **84.1** |

pute the prototypical representations following Equation (1). If the nearest representation of a test image is labeled as a synthetic class, the image will be considered fake.

## 4. Experiment

In this section, we first introduce the experimental setups and then provide a comparison between FSD and previous studies. We also visualize the feature space to highlight the superiority of our approach.

### 4.1. Datasets and Evaluation Metrics

**Datasets.** We evaluate our proposed method on the widely used GenImage dataset (Zhu et al., 2023), which contains $1,331,167$ real images from ImageNet (Deng et al., 2009) and $1,350,000$ generated images from 7 diffusion models and one GAN. The diffusion generators include Midjourney (mid, 2022), Stable Diffusion V1.4 (Rombach et al., 2022), Stable Diffusion V1.5 (Rombach et al., 2022), Wukong (wuk, 2022), ADM (Dhariwal & Nichol, 2021), GLIDE (Nichol et al., 2022) and VQDM (Gu et al., 2022). The only GAN utilized is BigGAN (Brock et al., 2019). The real images are first divided into 8 subsets. Each subset is subsequently partitioned into a training part and a test part. Each generator is associated with one specific subset and uses the category labels of the real images within the subset to produce synthetic images. Consequently, the GenImage dataset consists of 8 fake-vs-real subsets for analysis.

In this study, the synthetic images in different subsets are categorized into distinct classes, while real images from all subsets are aggregated into a single class. We also observe that 3 diffusion models, SD v1.4, SD v1.5 and Wukong, share an identical model structure, making it challenging to differentiate among them. Thus, we merge their corresponding subsets into a unified subset, named Stable Diffusion (SD). Ultimately, the dataset used in our experiments comprises 7 classes of images from different sources.

**Evaluation metrics.** Following previous AI-generated im-

age detection studies (Zhu et al., 2023; Wang et al., 2023; Luo et al., 2024), we adopt the accuracy (ACC) and average precision (AP) as our evaluation metrics. The threshold step for computing AP is set to $0.1$.

### 4.2. Implementation Details

To be more comparable with previous works, we adopt the ResNet-50 (He et al., 2016) pretrained on ImageNet (Deng et al., 2009) as the backbone of our model, which outputs a prototype vector of 1024 dimensions. Following Wang et al. (2023), the input images are first resized to $256 \times 256$ and then randomly cropped to $224 \times 224$ with random horizontal flipping during training. In contrast, only a center crop to $224 \times 224$ is performed after resizing the images during testing. The distance in the metric space is measured with Squared Euclidean Distance.

To fit the classification task, we employ different few-shot strategies for training and testing. Specifically, we select one synthetic subset as the test set and utilize the remaining 6 subsets to train a classifier. At every training step, 3 classes are randomly selected from the training set, with 5 samples chosen from each class for the support set and another 5 samples per class for the query set. We employ Adam as the optimizer to minimize the cross-entropy loss with a base learning rate of $10^{-4}$. We also adopt a StepLR scheduler with $\gamma = 0.5$ and $step\_size = 80000$. Each classifier is trained for $200,000$ steps, with a batch size of 16. During testing, we conduct a binary classification task by comparing the selected test class against the real class. For zero-shot detection, we randomly select 1024 samples from each training subset to compute the metadata vectors. Our method is implemented with the PyTorch library and all the experiments are conducted on a single A100 with 40GB memory.

### 4.3. Comparison to Existing Methods

We compare FSD with several state-of-the-art synthetic image detection methods and summarize the results in Table 1.

*Table 2.* Results of 10-shot cross-generator image classification on different subsets. Each classifier is trained on the dataset excluding the subset from the first column. All classifiers are tested on the 6 synthetic classes, reporting accuracy/average precision (%).

| Excluding Subset | Test Subset | | | | | |
|---|---|---|---|---|---|---|
| | Midjourney | GLIDE | ADM | SD | VQDM | BigGAN |
| Midjourney | 80.9 / 84.6 | 99.9 / 99.9 | 98.5 / 99.3 | 97.1 / 98.7 | 99.5 / 99.9 | 88.0 / 92.9 |
| GLIDE | 86.8 / 89.9 | 97.1 / 98.0 | 97.9 / 98.9 | 97.1 / 98.8 | 99.2 / 99.7 | 91.9 / 97.1 |
| ADM | 87.6 / 91.8 | 99.8 / 99.9 | 79.2 / 83.8 | 94.8 / 97.2 | 98.8 / 99.4 | 91.0 / 96.1 |
| SD | 86.1 / 89.7 | 99.9 / 99.9 | 97.4 / 98.8 | 88.8 / 92.5 | 96.6 / 98.5 | 89.5 / 95.4 |
| VQDM | 82.4 / 85.9 | 99.9 / 99.9 | 97.3 / 98.6 | 95.6 / 98.0 | 76.2 / 79.4 | 83.5 / 89.1 |
| BigGAN | 88.9 / 91.6 | 99.9 / 99.9 | 98.3 / 99.3 | 98.1 / 99.3 | 96.4 / 98.3 | 82.2 / 86.8 |

The compared group includes CNNSpot (Wang et al., 2020), Spec (Zhang et al., 2019b), F3Net (Qian et al., 2020), Gram-Net (Liu et al., 2020), DIRE (Wang et al., 2023) and LARE[2] (Luo et al., 2024). The first four methods are model-agnostic, focusing on identifying synthetic artifacts within fake images. The last two methods leverage the reconstruction error of Stable Diffusion to differentiate between fake and real images. To conduct a comprehensive comparison, we train 6 classifiers on different training sets, each of which excludes a specific test subset. However, previous studies typically provide several classifiers, and each classifier is trained on one distinct subset and evaluated on the others. To minimize the difference between our method and previous works, we report the average performance of 5 classifiers trained on non-test categories and evaluated on each test subset, which can better represent their generalization ability on each class.

As illustrated in Table 1, our FSD attains state-of-the-art performance across 5 out of the 6 test classes, surpassing previous state-of-the-art by +11.6% accuracy on average. The suboptimal performance of our model on the VQDM class can be attributed to the involvement of image quantization in VQDM, which significantly differs from the other generative models in our experiments. Another reason is that LARE[2] is specifically designed to detect synthetic images generated by diffusion models, which enables it to achieve unexpectedly high performance on the VQDM class. Although our few-shot method may not be directly comparable to other approaches due to accessing only a few samples in the test class, we still regard this discrepancy as a significant strength of our model. The definite improvement in performance demonstrates FSD's capability to effectively leverage a limited number of unseen samples.

Based on Table 1, we find that FSD achieves notable performance in zero-shot scenarios, in which the prototypical representations are calculated from samples in the training classes. These results show that our approach is also suitable for detecting synthetic images even when samples from the same sources are not provided, thus broadening the application scope of FSD. We attribute this advantage to FSD's ability to capture not only the intra-class characteristics but also the common features across all classes. Another observation is that the discrepancy between zero-shot results and few-shot results remains notable, which indicates the necessity of gathering training data from a wider range of generative models.

### 4.4. Cross-generator Classification

We evaluate the 10-shot performance of FSD on both seen classes and unseen classes. We also train 6 classifiers for the cross-generator image classification task, in which each classifier is trained on 6 out of 7 subsets in the dataset and subsequently tested on the remaining one. We report the accuracy and average precision across each scenario and summarize the results in Table 2.

As shown in Table 2, each row presents the classification results for one classifier across all the 6 synthetic classes. The results show that it is easy for a classifier to detect fake images from those classes encountered during training, which is consistent with the previous studies (Zhu et al., 2023; Luo et al., 2024). When it comes to detecting the unseen classes, as shown in the diagonal of Table 2, FSD suffers from a slight decline in performance, demonstrating the generalization capability of our model. Another finding is that FSD performs nearly perfectly on the VQDM class when this class is included in the training set. However, the performance significantly declines when this class has been excluded. This fact shows that it is still challenging to generalize across vastly different unseen generative models, indicating that our approach remains inadequate in capturing the discriminative features between classes.

We also observe the varying performance of FSD across different classes. For instance, images generated by GLIDE, ADM, SD and VQDM are relatively easy to detect by our model when these classes are included in the training set, while those from Midjourney and BigGAN are a little challenging to identify. Particularly, images from the GLIDE class can be distinguished with exceptional accuracy, even when not included in the training set. These facts indicate the necessity of treating images from different sources as

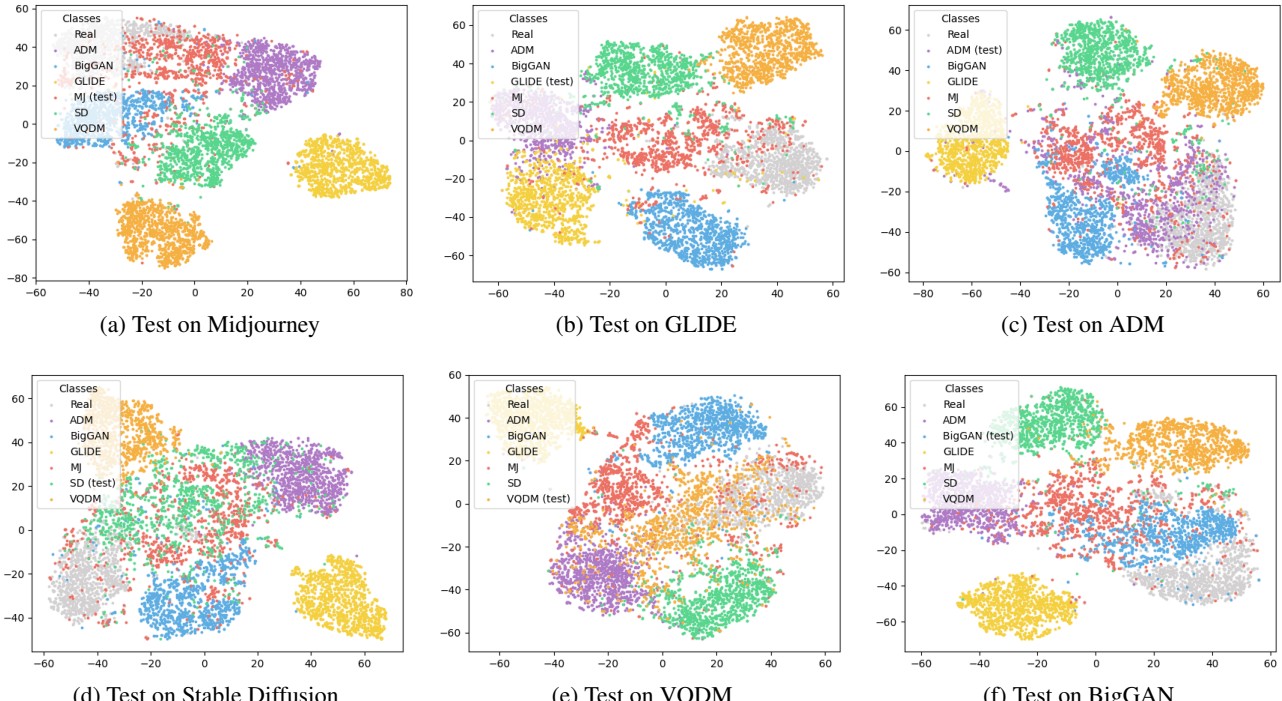

(a) Test on Midjourney       (b) Test on GLIDE       (c) Test on ADM

(d) Test on Stable Diffusion       (e) Test on VQDM       (f) Test on BigGAN

*Figure 4.* Visualization of network features for different classifiers over different classes using t-SNE. Each classifier is trained on those subsets excluding the test one in each sub-figure. The features are extracted from the final layer of our model.

distinct categories, which has not been fully explored in previous research. Moreover, it is essential to collect images from representative generative models for training, as generalization on similar generators is easier than on those that are vastly different.

### 4.5. Visualization

To further analyze the effectiveness of FSD, we utilize t-SNE (van der Maaten & Hinton, 2008) visualization to illustrate the feature space of classifiers on different training sets, as shown in Figure 4. Each classifier is trained on those subsets excluding the test one in each sub-figure. We randomly selected 1024 samples from the test part of each class for visualization. The features are extracted from the final layer of our model.

From Figure 4, we can see that features of images from different generative models exhibit distinct distributions, each of which is marked by a unique color. Most classes within the training set can be well-separated due to the large margins among their distributions. However, the boundary of the Midjourney class represented in red is not as clear as that of the others. This can be attributed to its API service, which may utilize multiple generative models for image generation as inferred from our analysis. Although the boundary of the test class in each sub-figure is a little ambiguous, the distribution of this class still remains iden-

tifiable, which demonstrates that our method successfully captures the intra-category features of unseen data.

The results also support the validity of our approach that categorizes samples from different generators as different classes. However, such categorization also introduces the issue of eliminating the significant differences between seen data and unseen data. As illustrated in Figure 4(e), the distribution of the VQDM class in orange exhibits a large overlap with that of the real class in gray, due to VQDM's significant differences in image tokenization compared to other models. This also indicates that our method still faces the challenge of detecting images from models that are significantly different from those generative models used in training, which will be the main focus of our future work.

### 4.6. Ablation Study

**Impact of the number of shots.** An important aspect is to determine the minimum number of samples required by our approach to achieve satisfactory performance. We choose different numbers of support samples in each class, ranging from 1 to 200. To present a comprehensive overview of this aspect, we conduct the experiment across all the 6 classifiers described in Section 4.4. We evaluate each classifier on the corresponding test class with varying numbers of support samples. To maintain a consistent total number of test samples across different settings, we set a fixed ratio of $1 : 3$

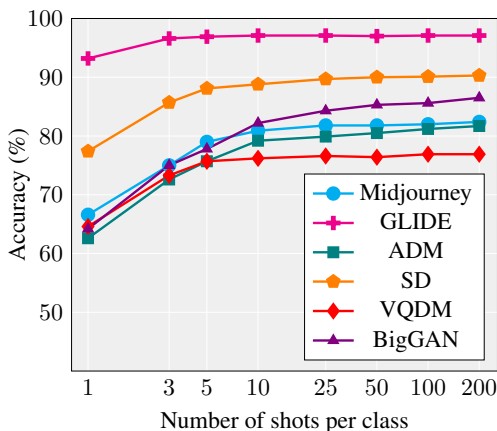

*Figure 5.* Influence of the number of shots. Each classifier is trained on those subsets excluding the test one. The 10-shot setting is the most cost-effective setting across all categories.

between the support set and the query set. The results are shown in Figure 5. Each line represents the performance of one classifier across varying sizes of support set.

From Figure 5, we can see that using more support samples results in better performance. However, it also increases the required computational resources. We find that detecting with 10 shots achieves an optimal balance between performance and resource consumption for most of the test classes. For example, our model brings the accuracy on the ADM class from 62.6% to 79.2% when increasing the number of shots from 1 to 10, with an improvement of +16.6%. However, it achieves 81.7% accuracy when using 200 shots, gaining only an increase of +2.5% over the 10-shots scenario. As it is easy to collect 10 samples from online generative models and the computational cost is very low, we ultimately decide to use the 10-shots results of FSD for comparison.

**Impact of the multi-class classification.** Different from previous studies that treat the fake images as a single class and train a binary classifier to detect them, we categorize the images from different generators as distinct classes. We argue that distinguishing images from different generators can also benefit the detection performance. Thus, we train another binary classifier and compare it with our method on the ADM class, which is difficult to distinguish according to our experiments. Both classifiers utilize the ResNet-50 as the backbone and both of them are trained on the dataset excluding the ADM class. We also apply the t-SNE visualization and randomly select 1024 samples for visualizing. Features of the binary classifier are extracted from the output layer of the ResNet-50, with dimensions of 2048. The visualization results are shown in Figure 6.

Figure 6(a) shows that our FSD successfully distinguishes those images across different classes which have been used in training. Images from the unseen ADM class colored

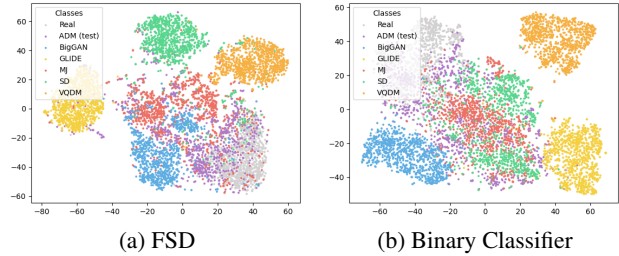

(a) FSD      (b) Binary Classifier

*Figure 6.* Visualization of network features for our FSD (a) and a binary classifier (b) on the ADM class. In the feature space of FSD, images from the unseen class in purple are tightly clustered together. However, they are significantly more dispersed in the feature space of the binary classifier.

in purple also cluster together in the visualization picture. In contrast, the traditional binary classifier fails to distinguish images from Midjourney and Stable Diffusion, as shown in Figure 6(b). The unseen samples in purple are quite dispersed, failing to form a compact cluster. Thus, distinguishing them from the real ones can be extremely difficult. This fact indicates that our method possesses a certain ability to extract intra-category commonality from unseen classes, thereby enabling an accurate differentiation among various classes. Moreover, we find that it is difficult for a classifier to distinguish samples from Stable Diffusion V1.4 and Stable Diffusion V1.5. One plausible explanation could be the shared usage of an identical backbone and VAE, which makes their products highly similar. This implies that we should carefully select the representative generative models to build the training set, which we leave for future work.

## 5. Conclusion and Limitations

The key point of detecting fake images in real-world scenarios is to make full use of the limited number of images from unseen generative models. In this paper, we propose the Few-Shot Detector (FSD), a novel AI-generated image detector based on few-shot learning. Experiments show that FSD achieves state-of-the-art performance in synthetic image detection with only 10 samples from the test generators, demonstrating the strong generalization ability of our model. We visualize the feature space to highlight the advantage of categorizing synthetic images by their sources, an approach that has not been widely adopted in previous studies.

However, our method still suffers from a few limitations. Since it is not feasible to obtain few-shot samples from all generative models, our collection of synthetic images is confined to part of these models, which may lead to a decrease in detection performance. Another weakness is that our method does not perform satisfactorily when detecting images from a vastly different generative model. We aim to address these issues in our future work.

## Acknowledgments

This work is supported by the National Science and Technology Major Project (2022ZD0116314), the National Natural Science Foundation of China (62106249, 62476070, 6243000159, 62102416), Shenzhen Science and Technology Program (JCYJ20241202123503005, GXWD20231128103232001), Department of Science and Technology of Guangdong (2024A1515011540), and the Natural Science Foundation of Jiangsu Province under Grant BK20243051.

## Impact Statement

AI-generated image detection has been a subject of extensive research. It is regarded as an important technology to reduce cheating, misinformation and harm on Internet platforms and social media. This work focuses on enhancing the performance of fake image detection rather than increasing synthetic data. All the images used in our experiments are from open-source datasets. We believe that our work can significantly contribute to the development of deepfake-detection applications in many real-world scenarios.

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
