# OpenReview forum: "Few-Shot Learner Generalizes Across AI-Generated Image Detection"
_ICML.cc/2025/Conference — ICML 2025 poster_

### Official Review · Reviewer_fx2j · 2025-03-12

**Overall Recommendation:** 1

**Summary:**

This paper adopts the concept of traditional few-shot learning (prior to 2022) and utilizes a prototype network to construct an AIGC image detector, with experimental validation demonstrating improved generalization performance.

**Claims And Evidence:**

Yes.

**Essential References Not Discussed:**

N/a

**Experimental Designs Or Analyses:**

The experiments in this paper lack testing on large-scale/high-quality datasets[i.e., WildFake[1], Chameleon[2]], and the compared methods are overly outdated.


[1] A Sanity Check for AI-generated Image Detection @ ICLR'25 https://arxiv.org/abs/2406.19435
[2] WildFake: A Large-Scale and Hierarchical Dataset for AI-Generated Images Detection @ AAAI'25

**Methods And Evaluation Criteria:**

Strength:

- This paper employs a prototype network to construct an AIGC image detector, reducing reliance on training data from newly generated algorithms.

Weakness：

- This paper merely applies ProtoNet without any task-specific improvements for AIGC detection. Therefore, I believe its academic contribution is quite limited.

- ProtoNet is also a decade-old algorithm, making it quite outdated in the few-shot learning domain. Recently, [https://dl.acm.org/doi/10.1145/3652583.3658035] utilized CLIP for few-shot AIGC detection. However, this paper does not provide a comparison with such approaches.

**Other Comments Or Suggestions:**

N/A

**Other Strengths And Weaknesses:**

N/A

**Questions For Authors:**

N/A

**Relation To Broader Scientific Literature:**

The authors applied an algorithm from the few-shot learning domain and introduced it into the AIGC detection task.

**Theoretical Claims:**

The paper does not present any theoretical contributions.

---

> ### Author Rebuttal · Authors · 2025-03-28
>
> Dear Reviewer fx2j,
> Thank you for your feedback and constructive comments. We appreciate the time and effort you invested in reviewing our work. Here are our responses to your concerns:
>
> Q1. Academic contribution is quite limited.
>
> To the best of our knowledge, our work is the first to systematically adapt few-shot learning for AIGC detection. While prior research has primarily focused on improving model generalization, our work uniquely leverages few-shot learning to mitigate performance degradation when detecting synthetic images from unseen domains or new generative models. The ability to achieve superior generalization with minimal samples from emerging generative models represents a key innovation of our approach.
> Specifically, our work makes the following key contributions:
> 1. Novel Adaptation of Few-Shot Learning: We propose a pioneering framework that leverages few-shot learning to significantly reduce performance degradation when detecting synthetic images from unseen domains or new generative models. This is particularly impactful given the rapid evolution of generative AI technologies.
> 2. Practical Efficiency: Unlike existing methods that require extensive retraining on new collected data, our approach achieves robust generalization with only minimal samples from new domains, offering a more scalable and resource-efficient solution.
> 3. Demonstration and Insights: Through comprehensive experiments, we demonstrate that our few-shot adaptation outperforms traditional fine-tuning approaches on widely used benchmarks.
>
> Furthermore, AIGC detection is a question-driven task, and our work provides a novel solution to address it. Our work introduces a new perspective in this field which can inspire future research directions and pave the way for more adaptive detection frameworks in the challenge of rapidly evolving generative technologies.
>
> Q2. ProtoNet is outdated in the few-shot learning domain.
>
> Our primary objective is to validate the fundamental effectiveness of few-shot learning. For this purpose, we intentionally adopt this vanilla yet classic method to ensure a comprehensive and interpretable evaluation. ProtoNet remains a principled and widely recognized baseline in few-shot learning, which allows us to isolate and rigorously assess the performance of few-shot learning without the confounding factors introduced by more complex tricks. Therefore, ProtoNet serves as an ideal choice due to its conceptual simplicity and proven reliability. This approach can help us better analyze the intrinsic advantages of few-shot learning, offering insights that may be obscured by more sophisticated but less concise methods.
>
> Q3. This paper does not provide a comparison with the given approach [1].
>
> We appreciate your insightful comment regarding the comparison with the approach presented in [1]. Our paper focuses on evaluating few-shot learning methods for AI-generated image detection, and we selected several widely recognized baselines for comparison, which have also been used by this work [2]. We will include comparisons with CLIP-based methods for a more comprehensive evaluation.
>
> Q4. Lack testing on large-scale/high-quality datasets.
>
> As emphasized in our paper, the primary goal of this work is to propose a novel few-shot learning framework for AI-generated image detection. We adopt the GenImage benchmark due to its widespread use in prior research, which facilitates direct comparisons with existing methods. While numerous datasets have been proposed for deepfake detection, few have gained broad recognition as foundational benchmarks. This study [2] presented at ICLR 2025 is too recent to be thoroughly validated in the literature. Additionally, our current evaluation is constrained by extremely limited computational resources. We appreciate this insightful suggestion and plan to include more experimental results in the supplementary materials.
>
> In conclusion, our endeavor is to introduce a novel few-shot learning application for AIGC detection, addressing the critical gap in generalization for rapidly evolving generative models. We sincerely appreciate your constructive feedback, which is helpful for us to identify key areas for further refinement. While we acknowledge the current limitations, we believe our study offers a new perspective in AIGC detection, and its methodological innovation—coupled with empirical validation—merits reconsideration for publication. We would sincerely appreciate it if you would reconsider the potential and novelty of our contribution.
> Thank you again for your time and insightful critique.
>
> [1] Sohail Ahmed Khan and Duc-Tien Dang-Nguyen. 2024. CLIPping the Deception: Adapting Vision-Language Models for Universal Deepfake Detection. In Proceedings of the 2024 International Conference on Multimedia Retrieval. Association for Computing Machinery, New York, NY, USA, 1006–1015.
>
> [2] Yan et al. A Sanity Check for AI-generated Image Detection. CoRR, abs/2406.19435, 2024.

---

### Official Review · Reviewer_N8UV · 2025-03-13

**Overall Recommendation:** 2

**Summary:**

The paper presents the Few-Shot Detector (FSD), an innovative approach to detect AI-generated images, particularly from unseen generative models. Traditional fake image detectors often struggle with generalization to new models due to the scarcity and high cost of collecting training data. FSD circumvents this challenge by reformulating the detection task as a few-shot classification problem, enabling it to effectively classify images based on limited samples from unseen domains.

## update after rebuttal
While I appreciate the authors’ rebuttal, key concerns remain unaddressed.  Therefore, I remain inclined to reject the paper.

**Claims And Evidence:**

Yes

**Essential References Not Discussed:**

No

**Experimental Designs Or Analyses:**

The comparison with state-of-the-art (SOTA) methods appears to be unfair. As previously mentioned, SOTA methods typically focus on single domain generalization, where models are trained on one type of generative model and tested on others. However, in line 290, the paper reports the average performance of six classifiers, all of which are trained on both GAN and DM models. This approach does not align with the standard practices of single domain generalization, thus skewing the results.

**Methods And Evaluation Criteria:**

Most existing methods address the domain generalization problem in forgery detection as a single domain generalization issue, primarily because generative models are not available for test images. This paper introduces few-shot learning into forgery detection, framing it as a multiple source generalization problem. While the proposed setting is theoretically reasonable, the paper does not adequately address key challenges associated with this new framework:

1.	The methodology lacks a clear strategy for obtaining test images from the same domain (i.e., generative models) in real-world applications.

2.	The paper does not establish a new benchmark for multiple domain generalization, which should include a comprehensive training paradigm, an evaluation setting, and a fair comparison with state-of-the-art methods.
Overall, while the paper presents an new perspective, it requires significant improvements to effectively address these critical issues.

**Other Comments Or Suggestions:**

No

**Other Strengths And Weaknesses:**

Strengths:

The attempt to formulate few-shot learning in forgery detection is a promising approach.

Weaknesses:

1.	The methodology lacks a clear strategy for obtaining test images from the same domain (i.e., generative models) in real-world applications.

2.	The paper does not establish a new benchmark for multiple domain generalization, which should include a comprehensive training paradigm, an evaluation setting, and a fair comparison with state-of-the-art methods.
Overall, while the paper presents an new perspective, it requires significant improvements to effectively address these critical issues.

**Questions For Authors:**

No

**Relation To Broader Scientific Literature:**

This paper attempts to formulate domain generalization in forgery detection using multiple sources. However, it falls short in establishing a coherent framework and lacks a robust evaluation strategy.

**Theoretical Claims:**

The nearest-neighbor method is employed to calculate the similarity between test images and prototypical representations. However, unlike the classic few-shot classification tasks that focus on image content, forgery detection emphasizes image authenticity. Consequently, a fake image may share similar content with real images, potentially leading to its misclassification as authentic.

---

> ### Author Rebuttal · Authors · 2025-03-28
>
> Dear Reviewer N8UV,
> Thank you for your detailed feedback and constructive comments. We have carefully considered each point you raised and would like to address them as follows:
>
> Q1: The lack of strategy for obtaining test images from the same domain in real-world applications.
>
> While current diffusion models are resource-intensive to train, widely adopted models are often accessible via APIs or open-source releases, serving as representative models. Our approach requires only a small batch of images from such a model to effectively detect generated content—not only from the same model but also across similar domains. Additionally, we have proposed a zero-shot detecting method in Section 3.3 to assess samples outside the training domain, ensuring robust generalization in real-world scenarios.
>
> Q2: Lack of a new benchmark for multiple domain generalization.
>
> Current widely used large-scale synthetic image datasets like GenImage and ForenSynths [1] have provided valuable resources for fake image detection, containing samples from diverse generative models with clear source annotations. We identify a critical gap in current research that most existing methods focus narrowly on binary classification (real vs. fake), overlooking the substantial domain-specific characteristics across different generative models. Our work serves as the first framework for multi-domain detection, and we have transformed GenImage dataset to a suitable benchmark for this task, laying groundwork for future benchmark creation once dataset diversity matures.
>
> Q3: A fake image may share similar content with real images.
>
> Our approach is based on the observation that synthetic images generated by different AI models exhibit distinct artifacts, as demonstrated in prior research [2]. While existing solutions often rely on training separate binary classifiers for distinct generative models—a process that is computationally expensive and impractical for large-scale deployment—our method takes a more efficient and generalizable approach.
> By focusing on classifying images based on their source generators rather than relying solely on content-based features, our network is explicitly trained to identify these model-specific artifacts. This design of our model ensures that it prioritizes forensic traces (e.g., noise patterns, texture inconsistencies, or spectral discrepancies) over semantic content, which is a key limitation in CLIP-based and other content-driven detection systems.
>
> Q4: This approach does not align with the standard practices of single domain generalization.
>
> Thank you for raising this important point. You're absolutely right to point out the divergence from conventional single domain generalization approaches. As detailed in our paper Section 4.2, this difference stems from our unique training paradigm which fundamentally requires diverse class samples during the training phase. We acknowledge that this represents a departure from standard practices, but it's a deliberate design choice that enables our model to learn more transferable features across domains.
> We anticipate that expanding the class diversity during training would further enhance the model's generalization capability, as it would allow the model to learn even more robust feature representations. However, this strategy may not generalize well to binary-classification tasks, where artifacts across categories can exhibit significant differences and shared characteristics among synthetic images may be absent. This is an important direction we plan to explore in future work. We appreciate this thoughtful observation and will carefully address this methodological distinction in our final version to ensure proper contextualization within the field of domain generalization research.
>
> Our approach pioneers a novel few-shot learning framework for deepfake detection and systematically explores the potential of multi-class detection in this field, demonstrating the feasibility of few-shot forensic analysis across generative models. While there exist limitations, it provides foundational insights for future research to address evolving synthetic threats. We sincerely appreciate your expertise in guiding methodological refinements, and we hope these contributions could be considered in the final assessment.
> Thank you for your time and understanding.
>
> [1] Sheng-Yu Wang, Oliver Wang, Richard Zhang, Andrew Owens, and Alexei A
> Efros. 2020. CNN-generated images are surprisingly easy to spot... for now. In
> Proceedings of the IEEE/CVF conference on computer vision and pattern recognition.
> 8695–8704
>
> [2] Liu, Fengyuan , et al. "Which Model Generated This Image? A Model-Agnostic Approach for Origin Attribution." European Conference on Computer Vision Springer, Cham, 2025.

---

> > ### Comment · Reviewer_N8UV · 2025-04-09
> >
> > Apologies for the delayed response. I appreciate the authors’ rebuttal, but I still have concerns in the following areas, which is why I am maintaining my original score:
> >
> > Q1: The lack of strategy for obtaining test images from the same domain in real-world applications.
> >
> > The authors emphasise that some current generative models (e.g., diffusion models) can provide a small number of samples via open-source implementations or APIs. However, in realistic scenarios, we may not be able to access any samples from these generative models at all, especially in the case of black-box models.
> >
> > Q2: Lack of a new benchmark for multiple domain generalization.
> >
> > The paper does not provide a clear comparison with existing SOTA methods under the same assumptions.
> >
> > Q3: A fake image may share similar content with real images.
> >
> > No experimental evidence is provided to support this claim.
> >
> > Q4: This approach does not align with the standard practices of single-domain generalisation.
> >
> > The authors did not include additional experiments to show whether their method remains effective under the same settings as SOTA methods (e.g., training on a single domain).

---

### Official Review · Reviewer_aJMC · 2025-03-19

**Overall Recommendation:** 4

**Summary:**

This paper introduces an approach to detecting AI-generated images by reframing the task as a few-shot classification problem. The Few-Shot Detector (FSD) uses a prototypical network to learn a specialized metric space, distinguishing between unseen fake images and real ones only using very few samples. By treating images from different generative models as separate classes and real images as a single class, FSD improves generalization to unseen models without extensive retraining.

**Claims And Evidence:**

Yes, the claims made in the submission are supported by evidence. The authors demonstrate through experiments on the GenImage dataset -- that FSD outperforms existing methods, achieving an average accuracy improvement of 7.4%. They provide analyses, including zero-shot and few-shot scenarios, cross-generator classification, and ablation studies on the number of support samples. Visualizations using t-SNE further support their claims.

**Essential References Not Discussed:**

Based on my knowledge, the paper discusses the essential related works necessary to understand the context and contributions. It cites prior studies on AI-generated image detection, diffusion models, GANs, and few-shot learning.

**Experimental Designs Or Analyses:**

Yes. The experiments are valid, both few-shot and zero-shot scenarios. The authors conduct cross-generator classification and ablation studies on the impact of the number of support samples.

**Methods And Evaluation Criteria:**

Yes, Reframing AI-generated image detection as a few-shot classification task is Using prototypical networks to learn a metric space is suitable for effectively distinguishing unseen fake images with limited samples. The use of the GenImage dataset as a benchmark and metrics like accuracy and average precision are standard and appropriate.

**Other Comments Or Suggestions:**

N/A

**Other Strengths And Weaknesses:**

Strength:
- reframing the detection task as a few-shot classification problem, which is a good contribution to the field.
- The use of prototypical networks is interesting in learning a specialized metric space that generalizes to unseen classes with limited samples.
- The method achieves state-of-the-art performance, with substantial improvements over existing approaches.
- Addressing the challenge of limited data availability from unseen generators makes the approach relevant for real-world applications.

Weakness:
- There is a notable performance gap between few-shot and zero-shot scenarios, indicating limitations when no samples from the unseen class are available.
- The approach assumes that images from different generators form distinct clusters in feature space, which may not hold true if generators produce very similar outputs.
- Some sections could benefit from clearer explanations, particularly the zero-shot classification approach and the differences between training and testing strategies.

**Questions For Authors:**

- How does FSD perform when faced with generative models that are significantly different from those in the training set, such as models using different architectures, styles, or data domains?

- In zero-shot scenarios, is there a way to improve performance without relying on samples from unseen classes? For instance, could domain adaptation or meta-learning techniques be integrated to enhance generalization?

- How sensitive is the method to the choice of support samples? Are there strategies for selecting the most representative or informative samples to enhance performance with minimal data?

**Relation To Broader Scientific Literature:**

The authors reconceptualized AI-generated image detection as a few-shot classification problem, building upon existing work in few-shot learning and AI-generated image detection. By employing prototypical networks, they extend methodologies used in few-shot learning to the domain of synthetic image detection. This approach addresses limitations in prior work that treated fake images as a single class and struggled to generalize to unseen generative models.

**Theoretical Claims:**

The submission does not present theoretical claims that require formal proofs. The methodology builds upon established techniques in few-shot learning, specifically prototypical networks. The contributions are primarily empirical, focusing on the application of these methods to AI-generated image detection.

---

> ### Author Rebuttal · Authors · 2025-03-28
>
> Dear Reviewer aJMC,
> Many thanks for your careful reading and valuable comments. We hope our reply further reduces potential misunderstandings.
>
> Q1. How does FSD perform when faced with generative models that are significantly different from those in the training set?
>
> Our comprehensive evaluation on the GenImage dataset has examined FSD's cross-generator generalization capability across diverse diffusion models with varying architectures, covering both pixel space and latent space image generation. The results provide valuable insights into the detector's robustness against structurally different generative models. While these initial findings demonstrate promising generalization performance, we acknowledge the need for further validation with real-world generative models and plan to expand our testing in future work to strengthen these conclusions.
>
> Q2. Is there a way to improve performance without relying on samples from unseen classes in zero-shot scenarios?
>
> We acknowledge the performance gap in zero-shot settings and agree that integrating domain adaptation or meta-learning could further improve generalization. In our work, we have observed that generative models with similar structures tend to cluster closer together in the learned metric space. This suggests that as we accumulate sufficient samples from representative model types, detecting images from fine-tuned or LoRA-trained models should become easier and more straightforward. We anticipate this collection of representative samples will help minimize the current performance gap in zero-shot detection.
>
> Q3: How sensitive is the method to the choice of support samples?
>
> Our observations indicate that with fewer than 5 samples, the results tend to fluctuate within a certain range, showing noticeable variability. However, as the number of samples increases to around 100 (which is typically not difficult to collect in practice), the performance becomes significantly more stable and reliable. This suggests that the method achieves robust and confident outcomes when provided with an acceptable number of support samples.
>
> In conclusion, our approach pioneers a few-shot-learning way for deepfake detection. We believe this direction holds potential as a future trend in the domain. We will revise the manuscript to include more details about the zero-shot classification and training strategy. Thank you for your consideration.

---

### Decision · Program_Chairs · 2025-05-01

**Decision:**

Accept (poster)

**Comment:**

This paper received very mixed reviews - accept, weak reject, reject.

The most negative reviewer, fx2j, did not engage in discussions or respond to the rebuttal. His review was very terse, providing limited justification for his negative rating. His concerns were that the selected few-shot algorithm was 10+ years old, and that the authors did not experiment on current datasets. The authors responded that they intentionally used an established few-shot algorithm to demonstrate the approach, and still achieved SOTA results. One of the suggested datasets is from ICLR 2025, presumably a submission under review, and therefore inappropriate to suggest. The other is from AAAI ‘25, published after the ICLR submission deadline. This review was largely disregarded by the AC.

Reviewer N8UV had some reasonable concerns, but others were not as significant to the AC. He was concerned about the requirement for a small number of sample images from representative domains, but this is not a difficult requirement and should be reasonably practical. Most methods assume something similar, explicitly or implicitly. He also asked for a benchmark for the multi-domain setting, but this seems extraneous as the paper’s goal is AIGC detection, not domain transfer. He believes the quantified comparisons did not test under the same settings, which the authors rebutted effectively. R2 responded to the rebuttal, reiterating a few points, which the authors then responded to.

R1’s review is very positive, and reasonably thorough. He appreciates the novel use of few-shot learning for this task, and the SOTA results. Overall the paper presents an interesting idea and should be a contribution to the community.